# Daily Habits of Brazilians at Different Moments of the COVID-19 Pandemic

**DOI:** 10.3390/nu14235136

**Published:** 2022-12-02

**Authors:** Tamires Cássia de Melo Souza, Juliana Costa Liboredo, Lívia Garcia Ferreira, Marina Martins Daniel, Laura Di Renzo, Francesca Pivari, Lucilene Rezende Anastácio

**Affiliations:** 1Department of Food Science, Faculty of Pharmacy, Universidade Federal de Minas Gerais, Belo Horizonte 31270-901, Brazil; 2Department of Food, Universidade Federal de Ouro Preto, Ouro Preto 35400-000, Brazil; 3Department of Nutrition, Universidade Federal de Lavras, Lavras 37200-000, Brazil; 4Section of Clinical Nutrition and Nutrigenomic, Department of Biomedicine and Prevention, University of Tor Vergata, 00133 Rome, Italy; 5Department of Health Sciences, University of Milan, 20142 Milan, Italy

**Keywords:** coronavirus, food, physical activity, well-being, screen use

## Abstract

Background: The COVID 19 pandemic impacted the health and well-being of different populations around the world. The aim of this study is to investigate the changes in the daily habits of Brazilians before and during two moments of the COVID-19 pandemic. Methods: A longitudinal study in which an online questionnaire (sleeping time, alcohol consumption, smoking, use of screen devices, physical activity, and dietary patterns) was applied at three moments. Results: The frequency of alcohol consumption, smoking, and sleep hours did not change significantly at different times. For the number of alcoholic beverages, there was a reduction in consumption from T0 to T1 and an increase from T1 to T2. There was a significant increase in hours of screen device use from T0 to T1, remaining high at T2. Finally, the level of physical activity in minutes reduced from T0 to T1, returning to base levels at T2. As for eating habits, there was an increase in the frequency of consumption of instant meals, fast food, and sweets at the first moment, with a significant reduction at the second moment. The consumption of legumes, milk and dairy products, bakery products, and meats was higher at T2. Conclusions: Some habits returned to or approached T0 levels. However, other habits remained unchanged, such as screen time and frequency of consumption of some food groups, throughout the last evaluation.

## 1. Introduction

The COVID-19 pandemic began in March 2020 [1], and in the middle of November 2022, more than 630 million cases worldwide had been reported—35 million in Brazil. Additionally, more than 6.3 million deaths had occurred, and Brazil ranked second, with more than 689 thousand deaths [2]. Due to the high transmission rate characteristic of the new coronavirus (SARS-CoV-2), numerous safety measures have been established to minimize the damage and exposure of the population, with recommendations for the use of protective masks, suspension of various activities, personal hygiene, and mainly, social distancing and isolation measures [3,4].

Staying at home was strongly suggested in several places around the world in order to contain virus transmission [5]. This significantly affected the health and well-being of several populations around the world, promoting changes in their dietary patterns [6,7,8], eating behaviors [9,10,11], physical exercise practice [12,13,14,15,16], mental health, and many other aspects such as sleeping quality, mood, and body weight [17,18,19,20,21,22].

In Brazil, studies have already demonstrated changes in the eating habits of adolescents [10], worsening quality of life for adults [15], worsening eating patterns in underdeveloped regions of the country [23], changes in the self-perception of the health of adult individuals [24], and high prevalence of psychiatric disorders and binge eating [25]. However, most studies carried out in Brazil and around the world showed a perspective on the changes observed at the beginning of the implementation of the lock-down decrees and distancing/isolation measures—more or less restricted according to each region—and so far, few have presented data collected at two or more moments, covering the periods of the flexibility of preventive measures [26,27,28,29,30].

The changes in daily habits occurring after the COVID-19 pandemic have been attributed to numerous factors [31]. Initially, the difficulty in accessing food was blamed on: The restriction of free movement to certain places of purchase; the closing of canteens and restaurants; the reduction in purchasing power due to the increase in unemployment rates; the change in the way of working, and consequently, the readjustment of daily habits; behavioral changes, fears, apprehensions and even the passing sensation of “vacation”; and economic changes and food insecurity [32,33,34,35]. Subsequently, changes in diet were also associated with the difficulty of changing habits acquired in the first phase of the lockdown [5]. Changing eating habits may have resulted in important changes in the nutritional status and overall health of the population [36]. Studies have also speculated about the possibility of a significant increase in chronic and nutritional diseases [37].

Therefore, it is essential to observe the context of the eating habits and lifestyle of the population during the COVID-19 pandemic, especially considering the different periods (including the gradual return of activities, the progress of vaccination campaigns, and changes in the economic and social scenarios). Such investigation can help to understand the implications of so many changes in the daily lives of individuals and support the implementation of new interventions and health strategies. Thus, the objective of the present study was to verify changes in Brazilian adults’ daily habits, lifestyle, and eating habits before and at two moments during the COVID-19 pandemic.

## 2. Materials and Methods

### 2.1. Study Design and Participants

This was a longitudinal observational study of data related to the daily habits of Brazilians during the COVID-19 pandemic. It included 464 individuals residing in any region of Brazil, over 18 years of age, who agreed to participate and answer the online questionnaire (Figure 1). The project was approved by the Ethics and Research Committee of the Universidade Federal de Viçosa (protocol number 35516720.5.0000.5153). All the proposed steps followed the guidelines provided by the Declaration of Helsinki [38].

The study’s first phase occurred from August to September 2020—approximately five months after the implementation of social isolation measures in the country. The volunteers were instructed to answer questions about their habits in the period before the pandemic (T0—retrospective data) and during the pandemic up until that time (T1). In Brazil, the measures implemented during that period included the suspension of non-essential activities (closing of restaurants, bars, malls, and gyms) and suspension of in-person classes at schools and universities, with the implementation of emergency online teaching [39,40,41]. For this study, the pre-pandemic period was defined as being from January to March 2020. The second phase of the study (T2) took place from May to June 2021—approximately ten months after the first phase of the research. The epidemiological scenario of the pandemic in Brazil at T1 and T2 can be seen in Figure 2.

### 2.2. Data Collection Questionnaire and Procedure

The semi-structured online questionnaire was based on other studies that had already been carried out during the pandemic [42,43] and were created on the Google Forms^®^ platform (Google Corp Inc., EUA, Menlo Park, CA, USA). The questionnaire link was spread through emails, reports, university websites, and social media (Facebook, Instagram, LinkedIn, and WhatsApp) using the snowball sampling technique [44]. The questionnaire was accessible via cell phone, computer, or any other device with an Internet connection. Filling out the form took about 15 min.

The first part of the research included the consent form. Participant responses were anonymous, and they could stop participating in the study at any stage before submitting the answers.

### 2.3. Variables

The variables collected were divided into three groups of simple and multiple-choice questions (Figure 3). The first group consisted of questions about gender (female, male, and others); age (in years); schooling (complete middle school, incomplete high school, complete high school, incomplete undergraduate degree, complete undergraduate degree, incomplete graduate degree, complete graduate degree); monthly per capita income (R$); physical/social distancing (full; partial; none); occupational status (unemployed; retired; working/studying thoroughly remotely; working/studying partially remotely; working/studying in person; others) and place of residence (city/state). Volunteers answered these questions at T1 and T2. The second group addressed daily habits and lifestyle questions, and the third group contained questions about eating habits, as described below.

### 2.4. Daily Habits

Regarding daily habits and lifestyle, participants were asked about sleeping time (hours), smoking habits, alcohol consumption (weekly and the number of standard drinks per occasion), screen time, such as the use of smartphones, computers, tablets, and TV (hours/day), and physical activity (minutes/week) at T0, T1, and T2. The frequency of consumption of alcoholic beverages was divided into six categories: Does not consume; rarely consumes; consumes once a week; consumes 2 to 3 times/week; consumes from 4 to 6 times/week and consumes every day (categorized, respectively, as 0; 0.5 times; 1 time; 2.5 times; 5 times and seven times). Smoking was verified and divided into five categories: does not smoke, smokes up to 10 cigarettes a day; smokes 11 to 20 cigarettes a day; smokes from 21 to 30 cigarettes a day, and smokes more than 30 cigarettes a day (categorized, respectively, as 0; 10; 15; 25 and 32 cigarettes/day). Screen time was divided into five categories: up to 4 h; from 5 to 8 h; from 9 to 12 h; from 13 to 16 h, and longer than 17 h (later categorized, respectively, as 3 h; 6.5 h; to 10.5 h; 14.5 h and 17 h). The practice of physical activity was divided into six categories: Does not practice; practices for less than 90 min; practices from 91 to 150 min; practices from 151 to 210 min; practices from 211 to 270 min and practices for more than 270 min per week.

### 2.5. Eating Habits

The eating habits evaluated included: The number of meals eaten at T0, T1, and T2 (breakfast, morning snack, lunch, afternoon snack, dinner, supper, and other meals); increase, reduction, or maintenance of the amount of food consumed, the habit of snacking between meals, use of delivery services and practices of cooking at home. In addition, the Food Frequency Questionnaire was applied at T0, T1, and T2. This questionnaire was adapted from the model proposed by the Food and Nutrition Surveillance System (SISVAN) [45] for the following groups: legumes; cereals; bakery products; milk and dairy products; fruits; meats; processed meats; vegetables; sugary drinks; instant meals; sweets and fast food. The frequency of consumption of food groups was reported as follows: Never, rarely, once a week, 2 to 3 times/week, 4 to 6 times/week, earlier, and more than once a day.

### 2.6. Data Analysis

Data were analyzed using the Statistical Package for Social Sciences^®^ software (SPSS^®^ Inc., Chicago, IL, USA) version 21.0. By using the Kolmogorov-Smirnov test, it was detected that the data did not have a normal distribution. Therefore, the results were presented as a median associated with the interquartile range and in frequencies. The Friedman test with Bonferroni correction was used to compare paired samples of numerical variables (T0, T1, and T2), while the Cochran’s Q test with Bonferroni correction was used for categorical variables. The McNemar test was used to compare the categorical variables at two different times (T1 and T2), and the Wilcoxon test was used to compare the numerical variables referring to the values obtained by the differences between T1 and T0 (ΔT1T0) and between T2 and T1 (ΔT2T1). The data were presented in tables, graphs, and figures. The significance level adopted was 5%.

## 3. Results

This research included 464 volunteers who participated in the two phases of data collection (Figure 1), and 80.4% (373) were from the southeastern region of Brazil, with a median age of 24 (32–40) years. Of the participants, 95.7% (444) had completed their undergraduate and graduate studies or were still studying to obtain such degrees. More than half of the sample consisted of professors and students, 19.6% (91) and 40.7% (189), respectively. Most of the sample was women, corresponding to 82.8% (*n* = 384) of the participants. There was less social distancing and changes in the occupational situation (more people studying/working in person) between T2 and T1 (Table 1).

Regarding daily habits and lifestyle, when comparing the three moments (T0; T1 and T2), it was possible to observe significant changes in the time spent in front of screens and devices (*p* < 0.001), in the number of drinks of alcoholic beverages per occasion. (*p* = 0.007), and in the weekly practice of physical exercise (*p* < 0.001) (Table 2). The median screen time increased from 6.5 h (3.0–6.5 h) at T0 to 10.5 h (6.5–14.5 h) at T1 and remained unchanged at T2. As for the number of drinks of alcoholic beverage, it was possible to notice a difference between all the times evaluated, with a decrease from T0 to T1 and an increase from T1 to T2. Finally, the practice of physical exercise was reduced by more than 30% in the first phase, returning to the baseline in the second phase.

Among the variables related to eating habits, 55.4% (257) of the volunteers reported having increased the amount of food consumed from T0 to T1, while this increase was reported by 36.0% (167) of participants from T1 to T2. Regarding the habit of using delivery services, the increase occurred for 50.2% (233) of the volunteers at T1 and for 48.5% (225) at T2. The habit of snacking increased for 48.1% (223) of participants at T1, and for 31.7% (147) at T2. The increase in home meal preparation was reported to be 66.2% (307) of the sample at T1, and by 41.6% (193) at T2. The reduction of the amount of food ingested, of snacking and meal preparation was significant when comparing T1 with T2 (*p* < 0.001), as shown in Figure 4.

Data referring to meals eaten in the period before the pandemic (T0) and at both moments during the pandemic (T1 and T2) demonstrated that the number of people having a morning snack and other non-traditional meals increased at T1. At T2, the number of people having a morning snack increased again while it decreased for other meals. These changes were statistically significant (*p* < 0.001 and *p* = 0.015, respectively), whereas there were no significant changes regarding the other meals—breakfast, lunch, afternoon snack, dinner and supper—when comparing the three moments (Figure 5).

By evaluating the frequency of food consumption, significant differences were observed in the legume group, milk and dairy products, meats, fast food, instant meals, sweets and bakery products (Figure 6). For the legume group, consumption remained similar between T0 and T1, with a significant reduction from T1 to T2 (*p* < 0.001). Consumption of milk and dairy products and bakery products also followed this tendency, that is, consumption of such products remained the same in the first phase of the research, and a decrease was observed in the second phase (*p* = 0.015 and *p* = 0.001, respectively). The frequency of meat intake remained constant from T0 to T1, showing an increase from T1 to T2 (*p* = 0.022). For the fast food and candy group, there was an increase in the first observation phase and a reduction in the second (*p* = 0.011 and *p* < 0.001, respectively). The consumption of instant meals increased from T0 to T1 and remained the same from T1 to T2 (*p* < 0.001).

## 4. Discussion

The research was carried out at two moments. The first questionnaire application took place approximately five months after the start of the social distancing measures in Brazil (August/September 2020). In this period, the scenario was considered to be a public calamity according to Legislative Decree No. 6 of 20 March 2020 and the recommendations included the interruption of services classified as non-essential—activities not necessary to guarantee the survival, health, supply and security of Brazilian citizens [46]. In the second phase of the survey (May/June 2021), numerous activities were already being resumed, such as gyms, bars and restaurants. Moreover, there was a hybrid return to schools in different regions, and resumption of several in-person activities and work. In this last phase, vaccination campaigns were at the very beginning and still uneven. Approximately only 12.0% of the Brazilian population had completed the vaccination protocol initially proposed at that time [47].

In Brazil, the context was different in the two phases of the application of the research. Initially, compliance with the physical and social distancing measures reflected concern about the epidemiological scenario and agreement with what was being implemented [48]. However, at the second moment of the research, the population was already exerting more pressure on the government to return to activities that were previously prohibited [49]—a pressure reinforced by political speeches that disagreed with the maintenance of restrictive measures [11]. The relaxation of restrictions affected some general characteristics of the participants of this study, evidencing significant changes in the practice of social distancing and the occupational situation/form of work during the COVID-19 pandemic. At T1, 60.3% of the volunteers reported to be fully compliant with the established measures. At the same time, at T2, that number dropped to 45.5%, showing an increase in the number of people who were partially complying with or were not adopting any type of social distancing (*p* < 0.001). A similar finding was observed in a study carried out with Brazilian adults, which showed an 11.1% reduction in participant compliance with the strictest physical/social distancing measures from the first phase (July 2020) to the second (February 2021) [28]. Regarding the occupational situation, the present study showed a reduction in the number of people working remotely and an increase in partially remote or entirely in-person work (*p* < 0.001).

The gradual return to in-person activities was accompanied by changes in the daily habits of the population. It was possible to notice that the practice of physical activity presented a significant decrease of more than 30.0% of the weekly exercise minutes at T1. Surveys worldwide have shown a similar pattern after social distancing measures began. For instance, almost 80.0% of a sample composed of 1613 adult Brazilians [50] and 48.9% of a sample consisting of 1491 Australian adults [51] reported negative impacts on physical exercise as a result of the COVID-19 pandemic [50]. There was also an average reduction of 40 min per week among the Middle Eastern population [52] and worsening or maintenance of sedentary levels in different locations [7,12,43,53,54]. Despite the reduction at T1 found in the present study, there was a significant increase in the practice of physical exercise at T2, where the median returned to baseline—from 80 min/week (0–120) to 120 min/week (0–180) (*p* < 0.001). It is worth mentioning that, even so, the weekly time spent exercising was below the new levels recommended by WHO (150 to 300 min per week of light to moderate physical activity) [53].

The time spent using screens and devices also changed at T1 compared to T2, with a median increase of 4 h daily. Nonetheless, at T2, even with the gradual return of in-person activities, screen time was still elevated and unchanged, remaining at the median of 10.5 h (6.5–10.5 h) daily. While almost 45.0% of the sample continued working/studying thoroughly remotely, it is speculated that screen use goes beyond professional reasons. In a study carried out with 725 Brazilians, 71.3% of the sample reported having increased the use of screens and devices to access media and the Internet, while 73.6% of the volunteers stated that online interactions promoted a sense of well-being [54]. These devices expand communication possibilities and have numerous attractive features, but, at the same time, when excessively used in the daily life of individuals, they can bring risks. Moreover, reducing the time spent in front of screens can be challenging after becoming a habit [55], especially in this sample, where 60.3% of the volunteers were teachers and students carrying out the activities remotely/online.

Some studies performed during the pandemic and in other situations have already shown that the increase in screen time has also been accompanied by changes concerning food and lifestyle, such as increased consumption of alcohol and smoking. However, the present study showed a significant reduction in the number of standard drinks of alcoholic beverages consumed in the first phase of observation (from T0 to T1), decreasing from 2.5 (0.00–2.50) drinks to 1.0 (0.00–2.50) drinks per occasion of consumption. There was also a significant change in the second phase, and the median consumption was 1.75 drinks (0.00–2.50)—still below the baseline. A possible explanation for the difference observed in the consumption of standard drinks of alcoholic beverages—since its frequency did not change—is that, at T1, safety measures included closing bars and restaurants, suspension of events, and restriction of the sale of alcoholic beverages in numerous contexts/environments. Although the offer of these services had already been resumed at T2, 45.5% of the volunteers still complied with total distancing, while 48.1% partially adhered to such measures. Furthermore, in many cultures, the intake of alcoholic beverages is a behavior directly linked to the context of socialization [56] and contact deprivation due to the pandemic may have contributed to the outcome found in the present study.

Some eating habits also vary due to the context experienced, such as several meals eaten, the amount of food ingested, the habit of snacking, and meal preparation. The reduction observed in the number of people having a morning snack and other meals, besides the most common ones, in the habit of snacking and in the preparation of meals can be justified by the gradual return of activities, as well as the resumption of the routine to something closer to “normal”—since there was a significant difference in the variables related to the way the volunteers worked. In the first phase of the study (T1), there were still many uncertainties regarding the COVID-19 pandemic, and the epidemiological and informative bulletins were still not able to clarify all doubts concerning the best forms of treatment, and deadlines for maintaining distancing measures, among other questions that bothered the population [57].

Therefore, people were looking for ways to adapt to the new routine, and the unpredictability of the duration and unfolding of the pandemic contributed to the idea that decisions regarding daily habits were also temporary and had exclusively immediate effects [58,59]. Thus, many food choices may have been made for different reasons, either to seek comfort due to stress, fear, or worries [6] or simply because of the feeling of “free time/vacation” that being confined could cause in some [60]. At T2, information about COVID-19 was more consistent, and the situation was no longer as unpredictable as before, including the return of previously suspended activities [57], which may have contributed to the resumption of eating habits. This pattern of changes was observed even in the results referring to the frequency of food consumption. There was a significant increase in the consumption of instant meals, fast food, and sweets from T0 to T1, but a substantial reduction from T1 to T2, returning weekly frequency to baseline. For the legume group, milk and dairy products, and bakery products, consumption remained constant in the first phase, significantly reducing in the second. For the meats, an increase in T2 was observed.

While it is possible to observe slight changes in eating habits in general, it is essential to search for and understand more complex factors that can significantly interfere with how individuals deal with food in the pandemic context, such as eating behavior and perceived stress. For instance, previous research has revealed that almost 50.0% of the sample assessed showed craving behavior (or food craving), especially for sweets, in the first phase of the study, and this was associated with numerous factors, such as socioeconomic status, lifestyle, eating habits and eating behavior [61].

The results found in the present study represented a situation contrary to what was observed in the first phase of this research [8], in which the general deterioration of the habits and lifestyle of the evaluated individuals was very clearly perceived. Nowadays, some habits have returned or come very close to the baseline findings, such as physical exercise levels and consumption frequency of ultra-processed foods (sweets and instant meals) and high-calorie foods (fast food). This tendency seems to accompany the idea that the initial and adverse effects of the pandemic on habits and quality of life can evolve positively, even if slowly, with the new perspectives of a routine closer to “normal” [30,62].

It is important to emphasize that the present study has limitations that should be discussed. The fact that the questionnaire was answered using a device with an Internet connection may have affected the scope of the research, as well as the loss of more than 60% of the initial sample from T1 to T2. The final sample was composed primarily of people who have attained higher levels of education and have elevated per capita income, and thus, it is not representative of all Brazilians. Therefore, it is essential to note that this work does not reflect the impacts of food insecurity afflicting the country, especially in the context of the pandemic and dismantling of public policies for social protection [35,63]. Furthermore, a study carried out with 16,440 Brazilian individuals showed that the perception of people on social isolation during the COVID-19 pandemic varies according to income and education [64].

Nonetheless, this research also has strengths, such as the fact that it was one of the pioneers in observing changes in the habits and lifestyle of the population during the COVID-19 pandemic in Brazil. The data allow us to understand that many changes have taken place and that a close look at these changes will be necessary. Other research that supports strategies to face new Public Health problems is also required.

Until now, it has been possible to conclude that daily habits and lifestyle, such as the frequency of alcohol consumption, smoking, and hours of sleep, did not show significant changes at the evaluated moments. The consumption of alcoholic beverages decreased from T0 to T1 and increased from T1 to T2. There was a significant increase in screen and device usage time from T0 to T1, which remained elevated at T2, even with the gradual return to in-person activities. Finally, the practice of physical activity reduced from T0 to T1 and returned to baseline from T1 to T2.

As for eating habits, there was a significant increase in the consumption of instant meals, fast food, and sweets at first, with a substantial reduction at the second moment. For legumes, milk and dairy products, bakery products, and meats, an increase in T2 was observed. Regarding meals, there was a reduction in the number of people having a morning snack at all observed moments, while there was an increase from T0 to T1 and a decrease at T2 concerning other non-traditional meals.

## Figures and Tables

**Figure 1 nutrients-14-05136-f001:**
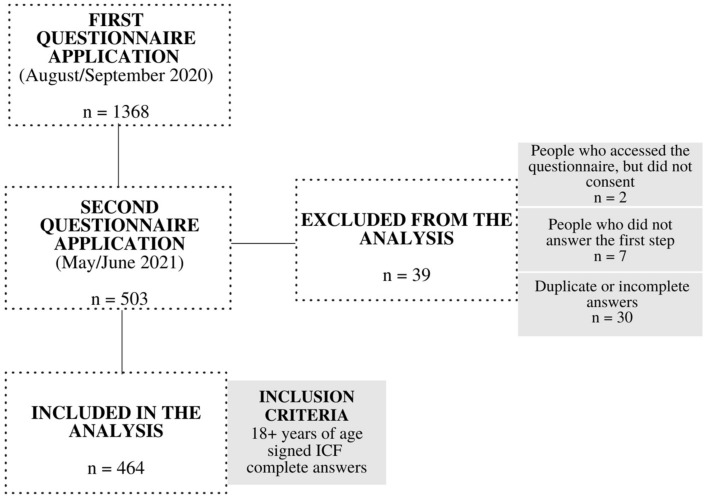
Recruitment of the research volunteers.

**Figure 2 nutrients-14-05136-f002:**
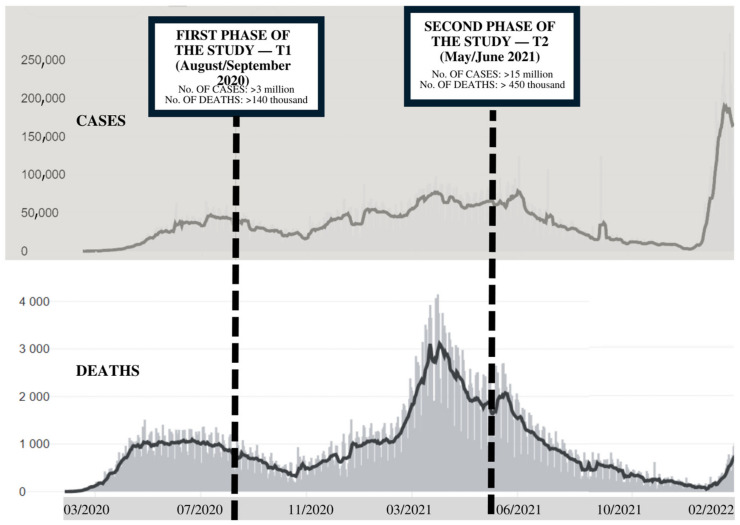
Research phases according to the epidemiological scenario of the COVID-19 pandemic in Brazil.

**Figure 3 nutrients-14-05136-f003:**
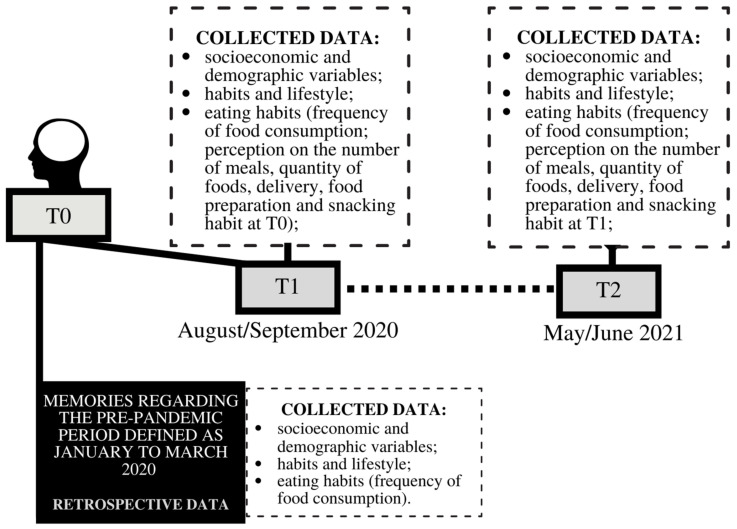
Detailing the periods and variables collected in each research phase.

**Figure 4 nutrients-14-05136-f004:**
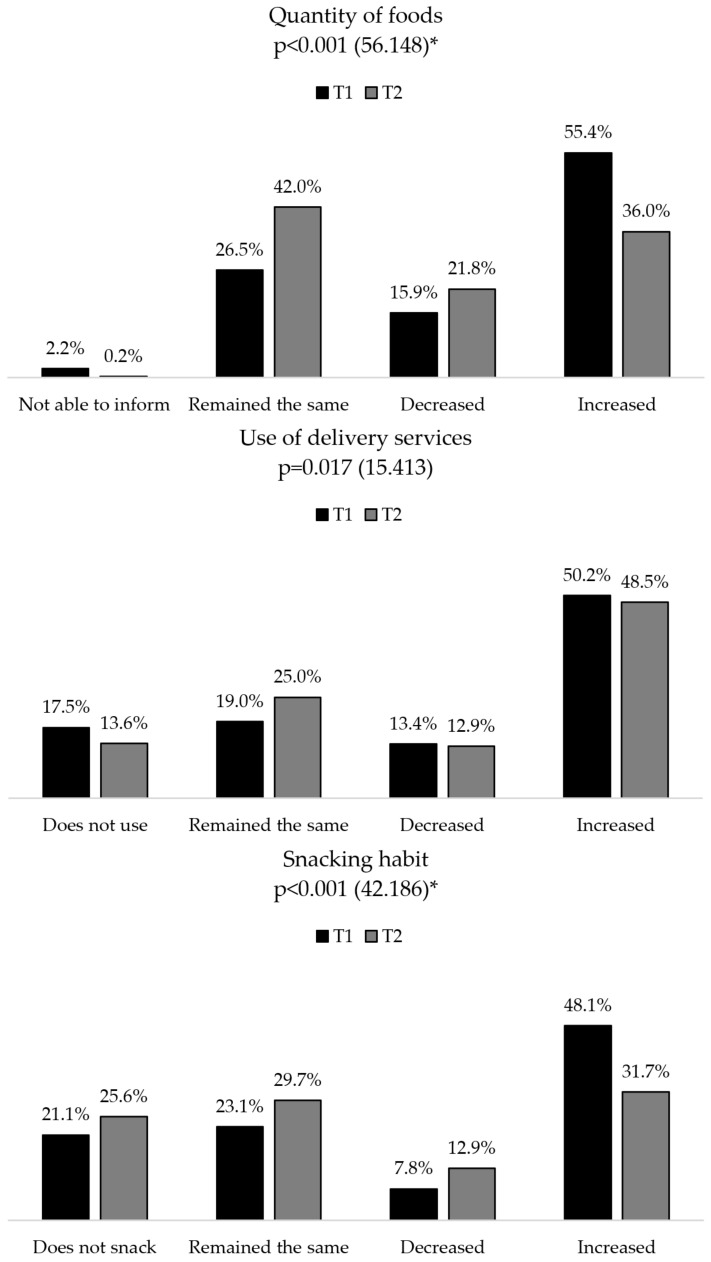
Changes regarding the amount of food ingested, use of delivery services, habit of snacking and meal preparation in the first and second phases of the research. *: *p* < 0.001.

**Figure 5 nutrients-14-05136-f005:**
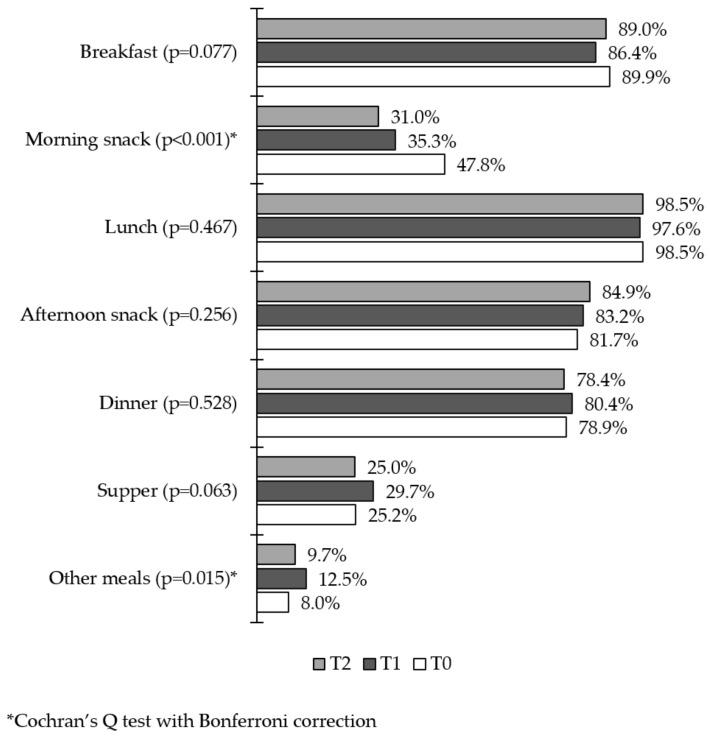
Daily meals in the first and second phases of the research.

**Figure 6 nutrients-14-05136-f006:**
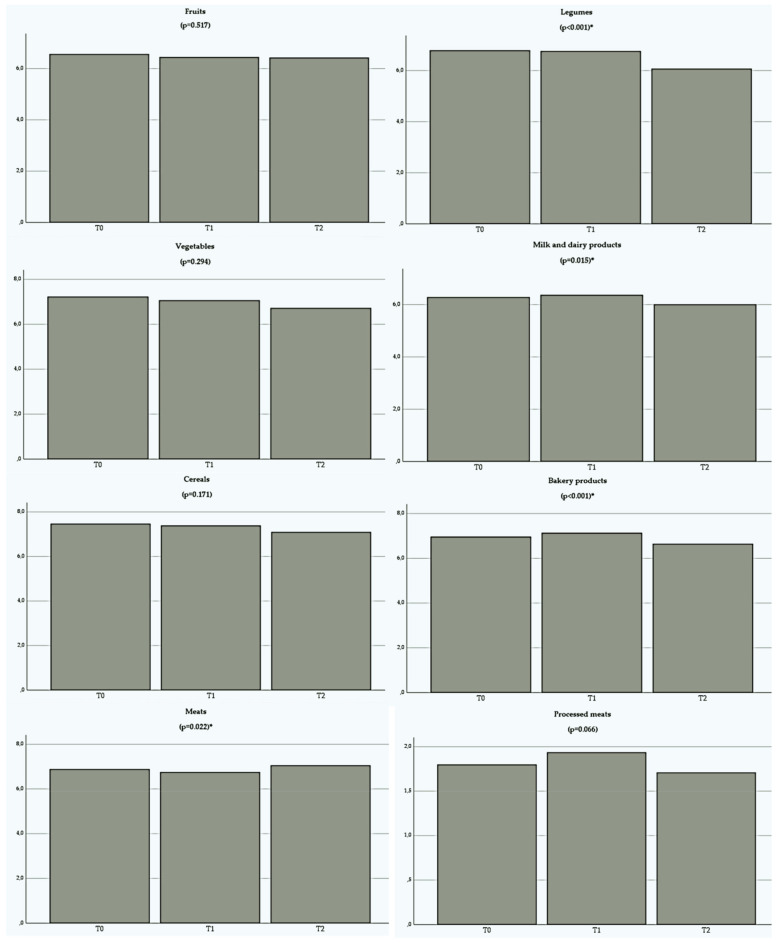
Frequency of Food Consumption reported by volunteers in the first and second phases of the research.

**Table 1 nutrients-14-05136-t001:** General characteristics of the volunteers who participated in the first and second phases of the research.

Variable	T1 Median (Q1–Q3) % (*n*)	T2 Median (Q1–Q3) % (*n*)	*p*-Value
Gender			
Female	82.8% (384)	82.8% (384)	-
Male	17.2% (80)	17.2% (80)	
Age (years)	24.0 (32.0–40.0)	24.0 (32.0–40.0)	-
Schooling			
Complete middle school	0.2% (1)	0.2% (1)	
Incomplete high school	3.7% (17)	3.7% (17)	
Complete high school	0.4% (2)	0.4% (2)	-
Incomplete undergraduate degree	19.8% (92)	19.8% (92)	
Complete undergraduate degree	23.3% (108)	23.3% (108)	
Incomplete graduate degree	41.6% (193)	41.6% (193)	
Complete graduate degree	11.0% (51)	11.0% (51)	
Monthly per capita income (R$)	4.702.5 (3.657.5–7.837.5)	4.702.5 (2.612.5–7.837.5)	0.779 ^1^
Social distancing			
Total	60.3% (280)	45.5% (211)	<0.001 ^2^
Partial	36.6% (170)	48.1% (223)	(44.982)
None	3.0% (14)	6.5% (30)	
Occupational status			
Unemployed	7.8% (36)	6.3% (29)	
Retired	3.4% (16)	3.9% (18)	<0.001 ^2^
Working/studying fully remotely	44.8% (208)	44.4% (206)	(40.503)
Working/studying partially remotely	27.8% (129)	22.8% (106)	
Working/studying in person	10.6% (49)	20.0% (93)	
Others	5.6% (26)	2.6% (12)	

^1^ Wilcoxon test; ^2^ McNemar-Bowker Test.

**Table 2 nutrients-14-05136-t002:** Daily habits before and during the COVID-19 pandemic.

Variable Median (Q1–Q3)	T0	T1	T2	*p*-Value
Screen time (hours/day)	6.50 ^a^ (3.00–6.50)	10.50 ^b^ (6.50–14.50)	10.50 ^b^ (6.50–10.50)	<0.001 ¹
Frequency of alcoholic beverage intake (times/week)	0.50 (0.00–1.00)	0.50 (0.00–2.50)	0.50 (0.00–1.00)	0.216 ¹
Number of standard drinks of alcoholic beverage (drinks/ocasion)	2.50 ^a^ (0.00–2.50)	1.00 ^b^ (0.00–2.50)	1.75 ^c^ (0.00–2.50)	0.007 ¹
Cigarette (number/days)	0.00 (0.00–0.00)	0.00 (0.00–0.00)	0.00 (0.00–0.00)	0.687 ¹
Sleeping time (hours/day)	8:00 (7:00–9:00)	8:00 (7:00–9:00)	8:00 (7:00–8:30)	0.067 ¹
Physical activity (minutes/week)	120 ^a^ (80–180)	80 ^b^ (0–120)	120 ^a^ (0–180)	<0.001 ¹

¹ Friedman test with Bonferroni correction (different letters indicate significant changes).

## Data Availability

Data is contained within the article.

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
