# Peer review of "Daily Habits of Brazilians at Different Moments of the COVID-19 Pandemic"

_nutrients, 2022, doi:10.3390/nu14235136_

Round 1
Reviewer 1 Report
Dear Authors, I have reviewed your manuscript entitled 'Daily Habits of Brazilians at Different Moments of the COVID-19 Pandemic ' I am glad to let you know that I enjoyed reading your paper. The paper is well-written and makes a novel contribution to knowledge and practice. However, there are some important issues that should be addressed to improve the quality of the paper. In the introduction section, a theoretical explanation of the mechanism of COVID-19 Pandemic impacts on Daily Habits is needed. Additionally, a discussion related to having food-induced changes in nutritional intake can be added. And regression or related methods such as PSM can be attempted to fully identify the differences between the two moments.
Reviewer 2 Report
The research paper aims of this study is to investigate the changes in the daily habits of Brazilians before and during two moments of the COVID-19 pandemic, through a longitudinal study.
The research methods are clear and was well conducted.
However, introduction need to be expanded.
Results, discussion and conclusion are in accordance.
Round 2
Reviewer 1 Report
The authors have made targeted changes to the previous suggestions and the quality of the manuscript has been further enhanced and improved. However, further improvements in the quality of the English language are still needed.